# How Widely are Supportive and Flexible Food Service Systems and Mealtime Interventions Used for People in Residential Care Facilities? A Comparison of Dementia-Specific and Nonspecific Facilities

**DOI:** 10.3390/healthcare6040140

**Published:** 2018-12-03

**Authors:** Rachel Milte, Clare Bradley, Michelle Miller, Olivia Farrer, Maria Crotty

**Affiliations:** 1Rehabilitation, Aged and Extended Care, Flinders University, GPO Box 2100, Adelaide 5001, South Australia, Australia; clare.bradley@sahmri.com (C.B.); maria.crotty@flinders.edu.au (M.C.); 2NHMRC Partnership Centre on Dealing with Cognitive and Related Functional Decline in Older People, Department of Rehabilitation and Aged Care, University of Sydney, Old Leighton Lodge, Hornsby Ku-ring-gai Hospital, Palmerston Road, Hornsby 2077, New South Wales, Australia; 3Institute for Choice, School of Commerce, University of South Australia, GPO Box 2471, Adelaide 5001, South Australia, Australia; 4Nutrition and Dietetics, Flinders University, GPO Box 2100, Adelaide 5001, South Australia, Australia; michelle.miller@flinders.edu.au (M.M.); Olivia.farrer@flinders.edu.au (O.F.)

**Keywords:** aged care, nursing homes, dementia, food, food services

## Abstract

While improved mealtime practices can reduce agitation, improve quality of life, and increase food intake for people in aged care, the degree of implementation of these strategies is unknown. This study describes food service practices in residential aged care facilities, focusing on units caring for people with dementia. An online survey was distributed to residential aged care facilities for completion by the food service manager (n = 2057). Of the 204 responses to the survey, 63 (31%) contained a dementia-specific unit. Most facilities used adaptive equipment (90.2%) and commercial oral nutritional supplements (87.3%). A higher proportion of facilities with a dementia-specific service used high-contrast plates (39.7%) than those without (18.4%). The majority of facilities had residents make their choice for the meal more than 24 h prior to the meal (30.9%). Use of high contrast plates (n = 51, 25%) and molds to reform texture-modified meals (n = 41, 20.1%) were used by one-quarter or less of surveyed facilities. There is a relatively low use of environmental and social strategies to promote food intake and wellbeing in residents, with a focus instead on clinical interventions. Research should focus on strategies to support implementation of interventions to improve the mealtime experience for residents.

## 1. Introduction

Poor nutritional status in people with dementia has been well described and is thought to result from reduced food and fluid intake below nutritional requirements, rather from an independent metabolic effect of the disease per se [1,2]. As the causes of malnutrition are not simply nutritional or physiological in nature but likely related to functional, cognitive, or behavioral symptoms, effective strategies are needed to address the multiple-factorial contributing factors for effective management [1,2,3,4,5]. To effectively address suboptimal food and fluid intake in people with dementia, residential aged care facilities need to consider not only the nutritional quality of the food, but also to address functional, cognitive, behavioral, psychological, and social barriers to adequate food and fluid intake [3].

The physical environment of the dining room can be used to promote independence in food and nutritional intake as well as the social and psychological wellbeing of residents, especially in those with dementia [2,6]. For example, the use of high visual contrast tableware (i.e., bright red crockery) can promote oral intake and functional independence by overcoming visual deficits [7]. Providing sensory cues such as the sight of tables set for dinner, sounds of meal preparation, or the smell of appealing foods cooking can orientate people with dementia to the fact it is mealtime [6]. Listening to music while dining has been shown to increase calorie intake in residents and, importantly, reduce agitation (which is experienced by as many as 93% of residents with dementia) [8,9]. Providing meals in a family-style arrangement (where people serve themselves from bowls of food at the table) rather than a plated service (where meals are served in a separate kitchen and transported to residents on trays) has been associated with improved nutritional intake, increased social interaction during meals, and increased independence [10,11]. Promoting choice at the time of the meal has been associated with increased quality of life in residents [12]. 

While these strategies have been shown to be effective, the studies in this area have their limitations. Predominantly they are smaller cohort studies or pre–post evaluations [2,13,14]. This is likely due to the difficulties associated with undertaking large-scale randomized controlled trials in this environment and context. There is also little information on the degree of implementation of the recommended strategies, and few studies of current practice in aged care food service have been published. Those undertaken have identified limited levels of choice of meals available to residents and the prominent use of traditional models of centralized food plating and distribution systems based around a main kitchen [15,16,17]. 

The aim of this study was to describe how food and dining is currently being provided in aged care facilities in Australia and to identify the extent to which supportive strategies and flexible food service systems are being implemented to improve food intake and wellbeing among people with dementia as well as those without dementia.

## 2. Methods

A web-based survey was distributed to residential aged care facilities across Australia to determine the characteristics of the food and nutrition support that they provided. The survey was open from March until June 2015 using SurveyMonkey^®^ (SurveyMonkey, San Mateo, California, United States of America). The study was approved by the Flinders University Social and Behavioral Research Ethics Committee (Project No. 6386). Informed consent was obtained prior to participation in the study. The survey results reported here are part of a larger survey that included 48 questions focusing on the characteristics of the facility and resident population, type of food service system used, and the timing and extent of choice available to residents with their meals. The current study reports the use of strategies to involve residents during mealtimes, support their independence and create a calm and optimal dining room environment, using techniques that have shown positive effects in previous systematic reviews and guidelines [6,18].

Facilities that identified themselves as containing a dementia-specific unit were then included in a dementia-specific subgroup for analysis. The facilities without a dementia-specific unit were included in an alternative subgroup for comparison. Where a facility included both dementia-specific and nonspecific units, they were included in the dementia-specific subgroup, using the rationale that the changes to the food service system that we were looking for would be applied similarly whether an entire facility or only one component of it was dementia-specific. A link to the questionnaire was emailed to facility contacts requesting that a food service manager or other relevant person complete the survey. Facility contacts were sourced from a commercially available database (AZGovBiz, http://azgovbiz.com.au) broadly representative of the total population of facilities in Australia, providing a list of administration, senior and middle management at residential aged care facilities nationally. The survey was sent to 2057 facility contacts covering all states of Australia, with a reminder email sent one month after the initial email. The survey was also advertised through relevant support organizations for food service managers. 

Data were analyzed using IBM SPSS, version 22 (IBM Corp. Released 2013. IBM SPSS Statistics for Windows, Version 22.0. Armonk, NY, USA). Pearson and Maximum Likelihood Ratio Chi Squared and Fisher’s Exact Probability tests were used to assess for differences in responses to categorical variables between those facilities with a dementia-specific unit and those without. 

## 3. Results

There were 204 complete responses to the survey (response rate of 9.8%), with 31% of the facilities reporting a dementia-specific unit. Table 1 summarizes the general characteristics of the facilities. Respondents were from all states of Australia. As expected, the majority of responses came from the three most populous states of Australia: New South Wales (34.8% of the survey responses), Victoria (22.5%), and Queensland (21.1%). Almost all (95.1%) of the facilities were catering for fewer than 200 beds. A little over half (57.3%) of the respondents were based in large cities in metropolitan or outer metropolitan areas while the remainder (42.6%) were based in rural centers or remote regions of Australia. Facilities in the dementia-specific category were more likely to be larger facilities catering for 100–199 beds (38.1%) or greater than 200 beds (9.5%) compared to the facilities in the non-dementia-specific group (100–199 beds = 18.4% and 200+ beds = 2.8%, respectively) (χ^2^ = 16.8, df = 4, *p* ≤ 0.001). The facilities in the dementia-specific group were also more likely than the non-dementia-specific group to be catering for a more heterogeneous mix of higher and lower care needs clients (dementia-specific group=81% vs. non-dementia specific group = 46.1%, χ^2^ = 20.2, df = 1, *p* ≤ 0.001). 

### 3.1. Food Service Systems in Place

Table 2 provides information on the food service characteristics. A large number (77.5%) of facilities were using a cook-fresh food service system (i.e., cooking meals on the day of service in the kitchen on-site) rather than using cook-chill or cook-freeze (7.8%) (where meals are cooked and pre-plated, stored, and then reheated for service) or bringing meals in from an external kitchen to the site (4.4%). Just over half (n = 115, 56.3%) of the respondents reported that meals were centrally plated and delivered to residents via trolleys. Most facilities were providing the ‘main’ meal of the day at lunchtime, with over half of respondents offering two hot and two cold choices for residents for the main meal. 

There were limited differences between facilities in the dementia-specific group and the non-dementia-specific group in the basic characteristics of the food service system (Table 2). A similar proportion of the dementia-specific group and the nonspecific group used a cook-fresh food system (76.2% vs. 78.0%), used a cook-chill or freeze systems (9.5% vs. 7.1%), used centralized plating systems (58.7% vs. 58.9%), plated food in the dining room from bulk-distribution carts (23.8% vs. 18.4%) and cooked meals in small kitchens accessible to residents (4.8% vs. 6.4%). However, the dementia-specific group were more likely to have the same number of hot and cold options at both lunch and evening meals than the non-dementia-specific group (43.5% vs. 23.7%) and less likely to have the lunchtime meal as the main meal (54.8% vs. 74.8%). 

### 3.2. Use of Supportive Mealtime Interventions

The frequency of different techniques used by facilities to improve meals and dining for residents who need special assistance or support with eating was analyzed (Table 3). Over 80% of surveyed facilities reported using adaptive equipment (n = 184, 90.2%) and commercial oral nutritional supplements (n = 178, 87.3%). Unpaid volunteers or family members provided feeding assistance in 42% (n = 87) of facilities. By comparison, high contrast plates (n = 51, 25%) and molds to reform texture-modified meals (n = 41, 20.1%) were used by no more than a quarter of facilities. 

There was a higher proportion of the dementia-specific group using high contrast plates than the nonspecific group (39.7% vs. 18.4%). However, for other techniques there were no significant differences between the two groups.

### 3.3. Support for Resident Choice and Flexibility

Table 4 shows the extent of choice in meal content and service size and the involvement of residents in mealtimes. Some (n = 63, 30.9%) facilities indicated that residents choose their meals at least 24 h before service, and a similar proportion reported that residents made their choice on the morning of the meal service (n = 59, 28.9%). Other facilities (n = 35, 17.2%) reported that they used a different system for resident meal selection; examples included residents placing orders a week or a few weeks in advance, facilities only offering a set menu or limited choices for meals, and kitchen staff selecting options for residents. One site reported requiring meal choices to be made the day before except for residents with dementia or memory impairment, who were asked at the time of serving. While a larger proportion of facilities with a dementia-specific unit had residents make their choice for the meal at the mealtime (25.4%) than facilities in the non-dementia-specific group (13.5%), this difference was not statistically significant (χ^2^ = 3.6, df = 1, *p* = 0.059). 

The majority of respondents were catering for at least one resident with a texture-modified diet, with 195 (95.6%) catering for resident(s) with a soft diet, 191 (93.6%) catering for minced and moist, and 196 (96.1%) catering for resident(s) with a smooth puree diet (Table 4). The number of choices offered for texture-modified diets for facilities with a dementia-specific unit and those without was compared (Figure 1). An increasing number of facilities reported they offered no choice for residents on a texture-modified diet as the level of texture modification increased. The proportion of facilities offering no choice for soft diet meals was similar regardless of dementia-specific status at both lunch and dinner, and for minced and moist diet at lunch. However, a higher proportion of facilities with a dementia-specific unit offered no choice to residents for minced and moist diets at dinner and for smooth puree diets at lunch and at dinner.

## 4. Discussion

This cross-sectional study identified that overall, food-service and mealtime practices were similar between aged care homes with a dementia-specific unit and those without. We had hypothesized that facilities with dementia-specific units may use more specialist strategies that had been evaluated with people with dementia, such as using decentralized food provision systems, serving (plating) meals in the dining room and allowing residents to choose their meal at the time of service, to compensate for the cognitive decline they could be experiencing and in accordance with available guidelines [6]. However, we found the adoption or uptake of these techniques to be limited. Instead, commercial supplements were among the most commonly reported techniques to support nutritional intake in the current study, despite research indicating that they are the least preferred strategy for family members of residents with dementia [19].

Oral nutritional supplements are just one way to address poor nutritional intake in frail older people, and there is growing evidence for other strategies that can be implemented for people with dementia [6,18]. Such strategies include multicomponent programs to improve the dining experience, incorporating a mixture of factors such as changing the physical environment and atmosphere of the dining room, changing food services systems to allow residents greater choice at mealtimes, and increasing staff involvement. These have been shown to maintain quality of life by increasing food intake, improving fine motor function, increasing body weight, increasing participation and communication at mealtimes and reducing food waste [10,11]. However, despite the growing evidence to support such practices, this study found very limited implementation of multicomponent strategies in the care facilities surveyed. There is limited information available regarding the extent of implementation of these strategies not only in Australia, but also internationally. One study in Norway found only limited involvement of residents (including those with and without dementia) in food service systems and planning—for example, none were involved in menu planning, and 90% did not participate in food preparation or meal provision (e.g. setting or clearing tables) [17]. In our study a low proportion of facilities indicated that residents were involved in preparation for meals (16.3% of facilities surveyed). In addition, in the Norwegian study most residents were unable to decide when they wanted to eat (76.3%) [17]. Similarly, in the current study 70% of facilities surveyed indicated that meals were served at a set time. By comparison, in Denmark there has been a push for traditional nursing homes to be converted or for new homes to be established as ‘Stay and Living Environments’ which have a greater focus on the social environment, and support greater involvement of residents in the planning and provision of their own meals—for example, involvement in planning the menu and preparing meals [20]. However, the majority of available evidence points to continuing widespread use of institutional-style and traditional food service systems internationally, with a focus on bulk production of food in a central kitchen by professional staff with limited choice, flexibility and involvement for residents [17,21]. 

We also hypothesized that facilities containing dementia-specific units would be using a greater variety of mealtime strategies to support nutritional intake than those without such units. However, overall, we identified no differences, with the exception of the use of high-contrast plates. Given the high prevalence of dementia among people living in residential aged care, a large number of people with dementia reside within the general residential care population, which could contribute to the similarity of service provision across these two groups. Nevertheless, the overall implementation of a number of the strategies that could benefit residents remains very low. There may be a number of reasons why these strategies are not implemented currently. It should be noted that access to dietitians (the recognized professional group with expertise in nutrition and dietary assessment and advice in Australia) within long-term aged care facilities is variable and often limited, with most being brought in consult on a ‘as needed’ basis to conduct menu reviews or assessment of individual residents [22]. Dietitians with experience and expertise in aged care would be able to provide advice regarding a variety of strategies to improve nutritional intake in residents, and plan for implementation in consultation with the care and food service staff of the facility. However, there are no existing guidelines or benchmarks for access to dietitians within aged care. Internationally there is evidence of low levels of access to dietitians in nursing homes, with less than half of German nursing homes surveyed having a dietitian available in one study, although there is evidence of much higher availability in the Netherlands (91.5% of facilities), Austria (83.5% of facilities), and Italy [21,23,24]. A recent study highlighted that while aged care cooks and chefs valued the input of dietitians they found poor access to dietitians a major barrier, either due to cost of engaging an external dietitian, difficulty in contacting them, and nonoverlapping working hours [25]. Cooks and chefs with improved access to dietitians reported more confidence in managing specialist diets for residents including for those with malnutrition in a previous study [25]. Cooks and chefs in aged care are likely to be time-poor, with much of their time focused on the day-to-day task of managing the food service system—a complex system which needs to provide food to meet the diverse needs of the residents at the right time within the resources currently available to them. Therefore, the limited use of many of the strategies included in the survey may reflect a lack of support for chefs and cooks in managing malnutrition within the current aged care system, and therefore over-reliance on the relatively simple strategy of providing commercial nutritional supplements. A limited choice of meals for residents requiring texture-modified diets, as well as the use of ‘leftovers’ from a previous day’s meal provided as a meal the following day (further reducing the variety and quality of meals provided), has been identified previously among residential care facilities [15]. People requiring texture-modified diets are known to be at higher risk of malnutrition, have a lower food intake which does not provide them enough of the key macro and micronutrients (including energy, protein, fiber, calcium and vitamin D) needed to meet their requirements [26,27]. Therefore, not offering a choice of main meal to residents on texture-modified diets could compound the already-negative effect of the diet on nutrient intake. 

There are some limitations to the current study which should be acknowledged. We used a commercially available database as our primary source of contact details for the sample in the study, so the representativeness of our sample of the total population was dependent upon the quality of this database. Moreover, our low response rate may have affected the representativeness of the resulting data. However, we were able to compare the basic characteristics of the responding facilities against figures from the Australian Institute of Health and Welfare (AIHW) [28]. We found the proportions of respondents from each of the eight states and territories in our study were similar to the proportions of facilities across Australia. There are also similarities between government-reported figures for facility size: 26% of the facilities in our sample catered for more than 100 beds compared to 21% of facilities nationally. 

## 5. Conclusions

Residential aged care facilities should provide quality care for people with dementia, including providing food and dining in a manner that respects the personhood of individuals [3,14]. This is the first study to our knowledge to compare food and dining between aged care facilities with a dementia specific unit and those without. Reported use of strategies to promote nutritional intake and support health and quality of life in residents was limited apart from oral nutritional supplements and adaptive cutlery, which were widely used. We were not able to gain in-depth information on why facilities did or did not implement particular strategies in this current study; however, this is area which is in great need of future research to assist with the translation of research and recommendations into practice. 

## Figures and Tables

**Figure 1 healthcare-06-00140-f001:**
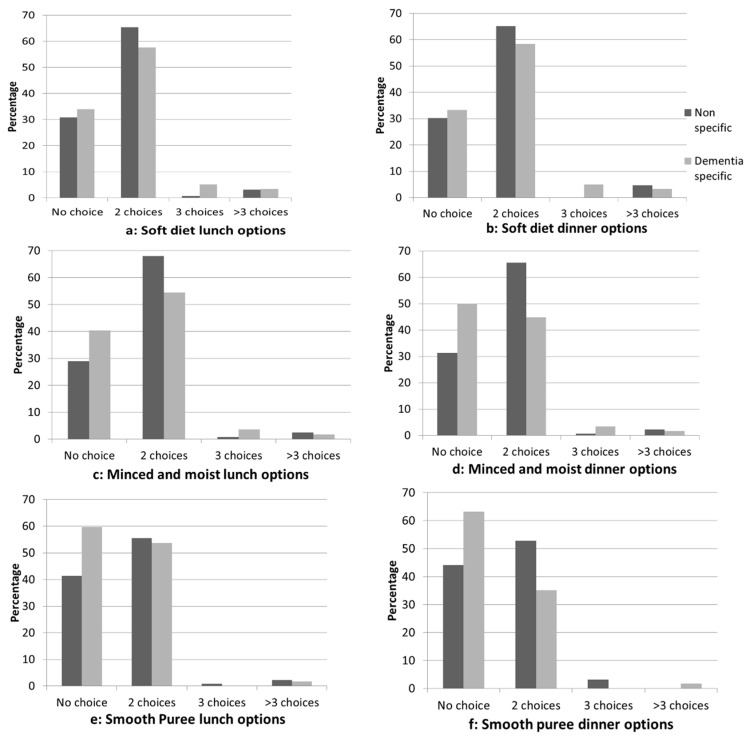
Comparison of the number of choices for texture modified diets in dementia-specific and nonspecific facilities.

**Table 1 healthcare-06-00140-t001:** Characteristics of responding facilities.

Characteristic	n (%)			Chi-Square
	Total	Not Dementia-Specific (n = 141)	Dementia-Specific (n = 63)	
State				
ACT NSW NT Queensland SA Tasmania Victoria WA	4 (2.0) 71 (34.8) 1 (0.5) 43 (21.1) 11 (5.4) 11 (5.4) 46 (22.5) 17 (8.3)	2 (1.4) 50 (35.5) 0 (0) 24 (17.0) 9 (6.4) 7 (5.0) 35 (24.8) 14 (9.9)	2 (3.4) 21 (33.3) 1 (1.6) 19 (30.2) 2 (3.2) 4 (6.3) 11 (17.5) 3 (4.8)	10.1, df = 7, *p* = 0.183
Location				
Metropolitan Outer metropolitan Rural or Remote	68 (33.3) 49 (24.0) 87 (42.6)	46 (32.6) 32 (22.7) 63 (44.7)	22 (34.9) 17 (27.0) 24 (38.1)	0.9, df = 2, *p* = 0.655
How many beds in this does this kitchen cater for?				
1 to 99 100 to 199 200 and above	144 (70.6) 50 (24.5) 10 (4.9)	111 (78.7) 26 (18.4) 4 (2.8)	33 (52.4) 24 (38.1) 6 (9.5)	16.8, df = 4, *p* ≤ 0.001
What type of facility is this ^1^				
Aging in place Dementia-specific Low care High and Low care High care Other	112 (55) 63 (31) 19 (9.3) 116 (57) 36 (18) 16 (8)	69 (48.9) 0 (0.0) 9 (6.4) 65 (46.1) 23 (16.3) 9 (6.4)	43 (68.3) 63 (100) 10 (15.9) 51 (81.0) 13 (20.6) 7 (11.1)	5.8, df = 1, *p* = 0.016 199.3, df = 1, *p* ≤ 0.001 3.6, df = 1, *p* = 0.058 20.2, df = 1, *p* ≤ 0.001 0.3, df = 1, *p* = 0.583 0.8, df = 1, *p* = 0.266

Abbreviations: ACT Australian Capital Territory, df Degrees of Freedom, NSW New South Wales, NT Northern Territory, SA South Australia, WA Western Australia, SD Standard Deviation ^1^ Participants were able to select more than one response to this question.

**Table 2 healthcare-06-00140-t002:** Food service characteristics of responding facilities.

Characteristic	n (%)			Chi-Square
	Total	Not Dementia-Specific	Dementia-Specific	
What is the main food service system used?				
Cook-Chill/Freeze Cook-Fresh Meals are brought in from an external company/kitchen Mixture of Cook-Fresh and Cook-Chill/Freeze	16 (7.8) 158 (77.5) 9 (4.4) 21 (10.3)	10 (7.1) 110 (78.0) 8 (5.7) 13 (9.2)	6 (9.5) 48 (76.2) 1 (1.6) 8 (12.7)	2.9, df = 3, *p* = 0.422
How are meals distributed to the residents?				
Bulk food plated in dining room Centrally plated Meals cooked in small kitchens accessible to residents Other	41 (20.1) 120 (58.8) 12 (5.9) 31 (15.2)	26 (18.4) 83 (58.9) 9 (6.4) 23 (16.3)	15 (23.8) 37 (58.7) 3 (4.8) 8 (12.7)	1.2, df = 3, *p* = 0.755
What is your current menu cycle length				
1 to 3 weeks 4 weeks 5 or more weeks	10 (5) 163 (81.1) 27 (13.5)	7 (5.0) 110 (79.1) 21 (15.2)	3 (4.8) 53 (85.5) 6 (9.7)	1.2, df = 2, *p* = 0.562
Is a seasonal menu offered at your facility?				
Yes No	169 (82.8) 32 (15.7)	115 (82.7) 24 (17.3)	54 (87.1) 8 (12.9)	0.3, df = 1, *p* = 0.567
When is the main meal?				
Both have the same number of hot and cold options Evening meal Lunchtime	60 (29.4) 3 (1.5) 138 (67.6)	33 (23.7) 2 (1.4) 104 (74.8)	27 (43.5) 1 (1.6) 34 (54.8)	7.9, df = 2, *p* = 0.019
How many hot choices are available at the main mealtime?				
1 2 3 or more No choice available	44 (21.6) 127 (62.3) 25 (12.4) 5 (2.5)	34 (24.5) 84 (60.4) 17 (12.2) 4 (2.9)	10 (16.1) 43 (69.4) 8 (12.9) 1 (1.6)	2.3, df = 3, *p* = 0.515
How many cold options are available at the main mealtime?				
1 2 3 or more No choice available	50 (24.9) 115 (57.2) 14 (7) 22 (10.9)	37 (26.6) 78 (56.1) 8 (5.8) 16 (5.8)	13 (21.0) 37 (59.7) 6 (9.7) 6 (9.7)	1.7, df = 3, *p* = 0.632
Are residents offered any of the following?				
Morning tea Afternoon tea Before bed snack Light refreshments available all day	196 (96.1) 196 (96.1) 184 (90.2) 138 (67.6)	136 (96.5) 136 (96.5) 126 (89.4) 95 (67.4)	60 (95.2) 60 (95.2) 58 (92.1) 43 (68.3)	0.0, df = 1, *p* = 0.704 0.0, df = 1, *p* = 0.704 0.1, df = 1, *p* = 0.730 0.0, df = 1, *p* = 1.000
Do you cater for the following special diets?				
Nourishing or High Energy High Protein Low fat Allergy meals (e.g. lactose free and gluten free) Low potassium/sodium Other	183 (89.7) 174 (85.3) 192 (94.1) 148 (72.5) 36 (17.6)	126 (89.4) 121 (85.8) 130 (92.2) 106 (75.2) 23 (16.3)	57 (90.5) 53 (84.1) 62 (98.4) 42 (66.7) 13 (20.6)	0.0, df = 1, *p* = 1.000 0.0, df = 1, *p* = 0.920 2.0, df = 1, *p* = 0.110 1.2, df = 1, *p* = 0.276 0.3, df = 1, *p* = 0.583

Abbreviations: ACT Australian Capital Territory, NSW New South Wales, NT Northern Territory, SA South Australia, WA Western Australia, SD Standard Deviation. ^1^ Participants were able to select more than one response to this question.

**Table 3 healthcare-06-00140-t003:** Use of supportive mealtime interventions in dementia specific and nonspecific facilities.

Technique	Facilities Indicating Use of Intervention n (%)			
	Total	Dementia Specific (n = 63)	Non-Dementia Specific (n = 141)	Chi-Square
Adaptive equipment (e.g. large handled cutlery or plate guards)	184 (90.2)	57 (90.5)	127 (90.1)	0.000, df = 1, *p* = 1.000
Commercial oral nutritional supplements	178 (87.3)	54 (85.7)	124 (87.9)	0.046, df = 1, *p* = 0.831
Redesigning menu to include resident favorite meals	138 (67.6)	46 (73)	92 (65.2)	0.872, df = 1, *p* = 0.350
Table cloths in dining room	136 (66.7)	41 (65.1)	95 (67.4)	0.026, df = 1, *p* = 0.872
Snacks available on demand	131 (64.2)	44 (69.8)	87 (61.7)	0.926, df = 1, *p* = 0.336
Involving family in feeding residents	123 (60.3)	36 (57.1)	87 (61.7)	0.212, df = 1, *p* = 0.645
Finger foods available on menu	121 (59.3)	40 (63.5)	81 (57.4)	0.433, df = 1, *p* = 0.511
Music during mealtimes	105 (51.5)	30 (47.6)	75 (53.2)	0.341, df = 1, *p* = 0.559
Use of volunteers during mealtimes	87 (42.6)	32 (50.8)	55 (39.0)	2.015, df = 1, *p* = 0.156
Staff joining residents for meals	53 (26.0)	17 (27.0)	36 (25.5)	0.002, df = 1, *p* = 0.964
High contrast plates	51 (25)	25 (39.7)	26 (18.4)	9.377, df = 1, *p* = 0.002
Using molds to re-form texture modified foods	41 (201)	16 (25.4)	25 (17.7)	1.152, df = 1, *p* = 0.283
Other	15 (7.4)	3 (4.8)	12 (8.5)	0.432, df = 1, *p* = 0.402
Resident helping with preparation for meals	33 (16.3)	12 (19.4)	21 (15.0)	0.320, df = 1, *p* = 0.572

**Table 4 healthcare-06-00140-t004:** Flexibility and extent of choice in food provision in dementia-specific and nonspecific facilities.

Characteristic	Facilities Indicating Use of Intervention n (%)			
	Total	Dementia-Specific (n = 63)	Non-Dementia-Specific (n = 141)	Chi-Square
When do residents choose the content of their meal?				
At the mealtime Morning of meal service Evening prior to meal service More than 24 h prior No choice provided Other	35 (17.2) 59 (28.9) 33 (16.2) 63 (30.9) 5 (2.5) 35 (17.2)	16 (25.4) 17 (27.0) 10 (15.9) 20 (31.7) 2 (3.2) 11 (17.5)	19 (13.5) 42 (29.8) 23 (16.3) 43 (30.5) 3 (2.1) 24 (17.0)	3.6, df = 1, *p* = 0.059 0.1, df = 1, *p* = 0.810 0.0, df = 1, *p* = 1.000 0.0, df = 1, *p* = 0.988 0.0, df = 1, *p* = 1.000 0.0, df = 1, *p* = 1.000
Which meal serve sizes are offered?				
Small Regular Large Choice of meal size provided at mealtime	128 (62.7) 160 (78.4) 123 (60.3) 103 (50.5)	45 (71.4) 52 (82.5) 45 (71.4) 32 (50.8)	83 (58.9) 108 (76.6) 78 (55.3) 71 (50.4)	2.4, df = 1, *p* = 0.119 0.6, df = 1, *p* = 0.442 4.1, df = 1, *p* = 0.044 0.0, df = 1, *p* = 1.0
Are residents able to access and use a kitchen area?				
Yes, also utilized for service meals Yes, separate to the kitchen used for serving meals No	32 (15.8) 120 (59.4) 50 (24.8)	12 (19.4) 31 (50.0) 19 (30.6)	20 (14.3) 89 (63.6) 31 (22.1)	3.3, df = 2, *p* = 0.194
Does the facility have a set time for meals?				
Yes, for all meals Yes, for lunch and dinner only No, all meals are offered at a range of times	143 (70.8) 50 (24.8) 9 (4.5)	39 (62.9) 18 (29.0) 5 (8.1)	104 (74.3) 32 (22.9) 4 (2.9)	4.1, df = 2, *p* = 0.131
Does you facility cater for any of the following texture modified diets?				
Soft diet Minced and moist diet Smooth puree diet	195 (95.6) 191 (93.6) 196 (96.1)	62 (98.4) 60 (95.2) 62 (98.4)	133 (94.3) 131 (92.9) 134 (95.0)	0.9, df = 1, *p* = 0.280 0.1, df = 1, *p* = 0.758 0.6, df = 1, *p* = 0.439

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
