# Peer review of "How Widely are Supportive and Flexible Food Service Systems and Mealtime Interventions Used for People in Residential Care Facilities? A Comparison of Dementia-Specific and Nonspecific Facilities"

_healthcare, 2018, doi:10.3390/healthcare6040140_

Round 1

Reviewer 1 Report

The presented work concerns the topic related to implementation of different mealtime interventions which can improve quality of life and increase food intake for people, especially with dementia, in aged care facilities. The methodology of work has been correctly planned. The authors used the online survey method and compared the obtained results using appropriate statistical methods.

In the discussion of the results, the authors tried to explain the rare use of guidelines and recommended strategies for serving and organizing meals in aged care facilities, especially with a dementia-specific unit. The observations and conclusions presented in this paper give an opportunity to assess the organization of nutrition departments in aged care facilities in Australia, but also allow for the formulation of practical recommendations to improve it. However, the discussion did not compare the organization of the nutrition and catering services in similar residential aged care facilities in other countries.

It is also worth to explain in more detail some of the comparison of results given in percentages of a given population given in the tables and text, e.g. in line 113, it is not clear to which group refer values given in brackets, in Table 1 and line 164/165 it is not known whether the declaration of the use of an elimination diet or a modified texture diet refers to the frequency of their use or only to the fact that such diets are used by at least one person in this institution.

Author Response

Reviewer 1: The presented work concerns the topic related to implementation of different mealtime interventions which can improve quality of life and increase food intake for people, especially with dementia, in aged care facilities. The methodology of work has been correctly planned. The authors used the online survey method and compared the obtained results using appropriate statistical methods.

In the discussion of the results, the authors tried to explain the rare use of guidelines and recommended strategies for serving and organizing meals in aged care facilities, especially with a dementia-specific unit. The observations and conclusions presented in this paper give an opportunity to assess the organization of nutrition departments in aged care facilities in Australia, but also allow for the formulation of practical recommendations to improve it. However, the discussion did not compare the organization of the nutrition and catering services in similar residential aged care facilities in other countries.

We thank the reviewer for this suggestion. We have added text to the discussion section providing a comparison between the food service system and access to dietitians covered in the current study and studies conducted recently internationally, to increase the relevance of the article to international readers. Please see additions in track changes. 

It is also worth to explain in more detail some of the comparison of results given in percentages of a given population given in the tables and text, e.g. in line 113, it is not clear to which group refer values given in brackets, in Table 1 and line 164/165 it is not known whether the declaration of the use of an elimination diet or a modified texture diet refers to the frequency of their use or only to the fact that such diets are used by at least one person in this institution.

We have added explanation to the brackets in the results section to make this clearer. We have also added wording to the results section to make it clearer that the results refer to that the diet is provided to at least one resident in the institution (i.e. not the frequency of their use within the population). 

Reviewer 2 Report

1) The article title is: “How widely are supportive  mealtime interventions used for people with dementia in residential  care facilities?”

However, the survey respondents were from both  dementia-specific group (31%) and non-dementia-specific group, so the  title has to be revised accordingly

 Line 69-71 needs to be revised as well as this study is not merely focused on “people with dementia”.

2) The title of this paper focuses on “how widely  are supportive mealtime interventions used…”, but Section 3.2 “Use of  supportive mealtime intervention” is relatively short (only two  paragraphs with six sentences in total). The title can  be re-framed to cover a wider scope that is covered in this study.

3) At the end of this paper, “Discussion” Section  is included, but without “Conclusion”. It is preferable to highlight/  summarise some key issues in the “Conclusion” section after “Discussion”  Section.

Author Response

Comments and Suggestions for Authors

1) The article title is: “How widely are supportive mealtime interventions used for people with dementia in residential care facilities?”

However, the survey respondents were from both dementia-specific group (31%) and non-dementia-specific group, so the  title has to be revised accordingly

We have revised the title accordingly as suggested to “How widely are supportive and flexible food service systems, mealtime interventions used for people in residential care facilities? A comparison of dementia-specific and non-specific facilities.”

 Line 69-71 needs to be revised as well as this study is not merely focused on “people with dementia”.

We have revised the aim of the paper as suggested to reflect more broadly the results presented as such “The aim of this study was to describe how food and dining is currently being provided in aged care facilities in Australia and to identify the extent to which supportive strategies and flexible food service systems are being implemented to improve food intake and wellbeing among people with dementia as well as those without dementia.”

 2) The title of this paper focuses on “how widely are supportive mealtime interventions used…”, but Section 3.2 “Use of supportive mealtime intervention” is relatively short (only two paragraphs with six sentences in total). The title can be re-framed to cover a wider scope that is covered in this study.

 We have revised the title as suggested to more broadly cover the wider scope of the paper to: “How widely are supportive and flexible food service systems, mealtime interventions used for people in residential care facilities? A comparison of dementia-specific and non-specific facilities.”

3) At the end of this paper, “Discussion” Section is included, but without “Conclusion”. It is preferable to highlight/ summarise some key issues in the “Conclusion” section after “Discussion”  Section.

 The last paragraph of the discussion contains the key issues and findings to highlight, and this has been relabelled as the Conclusion to make this clearer to the reader. 

Reviewer 3 Report

An interesting and well-written manuscript. It was sad that you received such a low response rate. It had a lot of value with more participating units.

There are some error in the reference list. 

1. Sometimes uppercase are used in the title of articles in journals, sometimes not.

Some journal's names are shortened, others are not. Please be consistent.

Author Response

3) At the end of this paper, “Discussion” Section is included, but without “Conclusion”. It is preferable to highlight/ summarise some key issues in the “Conclusion” section after “Discussion”  Section.

 The last paragraph of the discussion contains the key issues and findings to highlight, and this has been relabelled as the Conclusion to make this clearer to the reader. 

Reviewer 3: An interesting and well-written manuscript. It was sad that you received such a low response rate. It had a lot of value with more participating units. There are some error in the reference list. Sometimes uppercase are used in the title of articles in journals, sometimes not.

We have edited the titles of journals in the reference list to make capitalization consistent. 

Some journal's names are shortened, others are not. Please be consistent.

We have adjusted the reference list so that all journal names are spelled out in full to be consistent.